# Adolescents’ Out-of-School Physical Activity Levels and Well-Being during the COVID-19 Restrictions in Greece: A Longitudinal Study

**DOI:** 10.3390/jfmk8020055

**Published:** 2023-05-04

**Authors:** Alexandros Lazaridis, Ioannis Syrmpas, Themistoklis Tsatalas, Charalampos Krommidas, Nikolaos Digelidis

**Affiliations:** Department of Physical Education and Sport Science, University of Thessaly, 42100 Trikala, Greece

**Keywords:** life satisfaction, subjective vitality, adolescents, pandemic

## Abstract

The present study aimed to identify the impact of social distancing measures on adolescents’ physical activity (PA) levels and well-being during the implementation of COVID-19 restrictive measures. There were 438 participants (207 boys and 231 girls), aged 12 to 15 years old (*M* = 13.5, *SD* = 0.55). They completed online questionnaires on well-being and PA in three waves (December 2020, February 2021, and June 2021). Correlation analyses were conducted to examine the relationship between well-being and PA variables in the three measurements. Additionally, separate three-way repeated-measures ANOVAs were conducted to capture possible differences in students’ moderate to vigorous physical activity (MVPA) levels, life satisfaction, and subjective vitality among the three measurements due to gender, age, and interaction between gender and age. A significant relation emerged between the MVPA variables and well-being. In all measurements, adolescents’ PA levels did not meet the World Health Organization (WHO) recommendations of at least 60 min per day in MVPA. Students’ MVPA levels, life satisfaction, and subjective vitality were significantly higher in the third measurement compared to the first and second ones. Moreover, significant differences emerged in life satisfaction and subjective vitality between boys and girls in the first and third measurements, respectively. The COVID-19 restrictions appeared to negatively influence adolescents’ PA and well-being. Policymakers aimed at facilitating adolescents’ well-being in a similar situation in the future should not adopt measures restricting the participation of adolescents in PA.

## 1. Introduction

Physical activity (PA), which is defined “as any bodily movement produced by skeletal muscles that results in energy expenditure” ([1]; p. 126), may lead to an array of benefits for people’s health [2,3]. A significant number of studies indicated that regular participation in PA is beneficial for the immune system, protects the body against cardiovascular diseases, and aids the prevention, treatment, and control of hypertension, diabetes, obesity, osteoporosis, and depression [4,5]. The World Health Organization (WHO), taking into consideration the aforementioned findings, recommended that children aged between 5 and 17 should be engaged in moderate to vigorous-intensity physical activities (MVPA) for at least 60 min every day [6]. However, the findings of a study articulated that children do not meet the WHO recommendation [4,7,8,9]. More specifically, in Greece, only 12% of children over 13 and 14% over 15 meet the aforesaid recommendation [10].

PA is a critical factor that can also influence children’s well-being [11,12,13]. PA has been associated with well-being by favouring mental health [14,15,16,17]. Adolescents’ participation in PA reduces depression [18], promotes higher self-esteem [19], and boosts their physical and mental health [20,21,22]. Even short periods of moderate PA can positively affect their mood [23].

Furthermore, interaction through PA plays a major role in enhancing social support [24], explaining why PA has been found to contribute to young people’s mental health [25]. People’s engagement with outdoor activities seems to be de-stressing and leads to health benefits [26,27]. Spending time in nature has been positively correlated with the management of mental health problems and the avoidance of anxiety and depression feelings [28]. Generally, engaging in outdoor activities enhances mental well-being [29,30] and, in particular, allows adolescents to develop feelings of life satisfaction [31,32,33].

According to the United Nations Agenda for Sustainable Development Goals, health goals will promote healthy living and well-being at all ages [34]. Priorities need to be set for children to obtain future health [35]. More specifically, a lot of attention should be paid to adolescents’ well-being [36] because adolescence is an essential period of life for human identification and attitude development to be followed in adulthood [37,38].

Due to global changes, well-being has increasingly been at the centre of various research studies, leaving some issues unanswered or unclear [39]. Well-being refers to the state of feeling pleasure, strength, and wellness due to a satisfactory life [40]. It is an essential life component that includes liveliness, vitality, and the capacity to recover quickly from difficulties [41,42]. When motivation is connected with interests that enhance satisfaction, then experiential activities highly affect the increase in well-being [43]. Well-being depends on psychosocial life satisfaction. An inclusive school environment can essentially support young people psychologically, thus improving their quality of life [44]. Positive well-being helps adolescents develop feelings of happiness and satisfaction, efficiently deal with social relationships, and have optimistic expectations [45]. Consequently, adolescents can contribute to their community more efficiently, which can have a positive impact in adulthood as well-being helps young people develop social and problem-solving skills, making them productive [46,47]. Thus, they can efficiently cope with difficulties and stressful situations while their social integration is promoted [48].

During the COVID-19 pandemic, governments had to implement social distancing measures aimed at preventing the spread of coronavirus. These social distancing measures included an array of closure policies (e.g., schools and workplaces) and stay-at-home policies (e.g., restrictions of public events, outdoor gatherings, and internal movement). Because of social distancing, the ability of people to move freely was confined. Thus, their life was seriously affected as face-to-face interaction decreased [49], and feelings of loneliness arose which resulted in negative psychological conditions [50]. The restrictive measures against the COVID-19 pandemic affected people’s psychosocial well-being [51]. Additionally, a significant number of parents reported that children’s and adolescents’ psychological health was affected [12]. The COVID-19 pandemic era has been strongly linked with a negative impact on young people’s physical health and quality of life [13,52,53]. For example, [54] revealed that loneliness and social isolation during the pandemic raised the possible risk of depression and anxiety in children and adolescents. Similarly, a systematic review by Nearchou and colleagues [55], with twelve studies and 12.262 participants, revealed that the COVID-19 pandemic was associated with increased levels of depression and anxiety in adolescents. However, the methodological quality of these studies was low to moderate [55].

Regarding gender and age differences on children and adolescents’ well-being during the COVID-19 pandemic, the literature review revealed ambiguous results, as some studies reported significant differences in well-being between gender or age, while some others did not find any effect of gender or age on their well-being during the pandemic [55,56].

Furthermore, closing learning institutions for an extended period decreased student MVPA levels, resulting in self-isolation that was detrimental to healthy lifestyles [7,57]. Additionally, due to the confinement measures, sedentary lifestyles were increased at the expense of overall PA [58,59]. The decrease in PA seems to have had a negative effect on the well-being of people who were more physically active [60]. The well-being of people who were not infected but were forced to comply with the measures as a precaution was negatively affected [61,62,63]. The above results were also confirmed by the cross-sectional study of Morres and his colleagues [13] who found that increased levels of PA were positively related to adolescents’ well-being, whereas sedentary behaviours were negatively related to their well-being during the COVID-19 pandemic.

Concerning school closures, substituting face-to-face activities with online services might have been challenging for some but many difficulties had to do with the isolation per se. Adolescents seemed to be highly stressed as their scheduled program had to be dramatically modified and the emotional strain to stay away from activities required social isolation [64]. Stress in adolescence is negatively related to the long-term stressful life of adolescents during adulthood [65]. A number of cross-sectional studies and systematic reviews showed a decrease in children’s PA during the stay-at-home time [12,66,67,68,69]. However, a recent review supported that the findings, regarding the link between children and adolescents’ PA and mental health during the COVID-19 pandemic, were equivocal [70]. More specifically, a number of studies reported a positive link between adolescents’ PA and mental health during the COVID-19 pandemic, while some others did not reveal a significant relation between adolescents’ PA and mental health during the pandemic [70].

The purpose of the present study was to examine the impact of three different restrictive measures (i.e., closed schools and closed structured outdoor activities, opened schools but closed structured outdoor activities, and opened schools and structured outdoor activities), that have been established by the Greek government during a semester on adolescents’ MVPA and well-being. Based on the above, we believe that the three different types of restrictions implemented in Greece during the COVID-19 pandemic will also have a different impact on children’s and adolescents’ PA levels and well-being. Thus, two research hypotheses guided the present study: (1) How or to what extent did the various restrictive measures affect the PA levels and well-being of adolescents in Greece? (2) Were there any gender and age differences in adolescents’ PA and well-being during the different restrictive measures implemented in Greece due to the COVID-19 pandemic?

## 2. Materials and Methods

### 2.1. Participants

The sample consisted of 438 students (207 boys and 231 girls). Their age ranged between 12 and 15 years old (*M* = 13.5, *SD* = 0.55). Participants had enrolled for the school year 2020–2021 in the seventh (*N* = 151), eighth (*N* = 141), and ninth (*N* = 146) grade of a junior high school in Athens, Greece. This school was purposefully selected because its students came from different areas of Athens and thus came with various social and economic backgrounds. The study was implemented with the approval of the Ethics Committee of the authors’ university (Protocol code: 1675; Date of approval: 7 October 2020). Additionally, parents’/guardians’ consent and students’ assent forms were obtained via email before data collection. Students voluntarily participated in the present study, and they were informed that they could withdraw their participation at any time.

### 2.2. Instruments

Physical Activity. Two items were delivered to assess students’ PA levels [71]. More specifically, one item was measuring students’ frequency to participate in out-of-school moderate to vigorous-intensity PA (MVPA; e.g., “Outside school hours: How often do you usually exercise in your free time, so much that you get out of breath or sweat?”) and their answers were given on a 5-point Likert scale ranging from 1 (once a month or less) to 5 (every day). The second item was assessing students’ amount of out-of-school MVPA (e.g., “Outside school hours: How many hours do you usually exercise in your free time, so much that you get out of breath or sweat?”) and participants answered on a 6-point Likert scale ranging from 1 (never) to 6 (about 7 h per week). Both items have been used in large epidemiological studies of the WHO’s Health Behaviour in Schoolchildren (HBSC) survey [72,73], have shown acceptable reliability and validity [71,74], and have already been used in the Greek language [75] to capture adolescents’ PA levels.

Well-being. Cantril’s life satisfaction ladder [76] and a short version of the subjective vitality scale [77] were used to measure students’ well-being during the COVID-19 pandemic. More specifically, Cantril’s life satisfaction ladder consisted of one item where participants indicated how satisfied they were at that point with their lives. Students’ responses were given on a 10-point Likert scale from 1 (worst possible life) to 10 (best possible life). Subjective vitality scale consisted of five items (e.g., “Over the past 3–4 weeks … I felt I had a lot of energy”) and participants answered on a 5-point Likert scale from 1 (strongly disagree) to 5 (strongly agree). Both Cantril’s ladder and subjective vitality scale have already been used in similar studies conducted in Greece [78,79]. In the present study, a confirmatory factor analysis (CFA) on the initial measure of the subjective vitality revealed acceptable goodness-of-fit indices: chi-square = 23.24, *df* = 4, TLI = 0.92, CFI = 0.97, RMSEA = 0.11, 90% RMSEA = 0.06 to 0.15 [80].

### 2.3. Procedure

Three waves of measurements were conducted in December 2020, February 2021, and June 2021. An online version of the instruments was developed by using Google Forms. During this period, different protocols had been established by the Greek government to ensure social distancing and prevent the COVID-19 virus spread. These protocols included measures that influenced students’ daily life (e.g., schooling, walking, leisure, and exercise, etc.).

More specifically, the first measurement took place in the third week of December 2020 when schools had already been closed for more than one and a half months. During this period, there was prolonged confinement at home as a total lockdown had been in progress, and specific restrictions were established to maintain social distancing. More specifically, moving outside the house was allowed only for necessary purposes. Citizens could walk/exercise outdoors only if they had sent a specific text message asking for permission. However, this could last for a limited amount of time and within a short distance from peoples’ permanent residence. Structured PA and mass gathering activities were forbidden. Additionally, people were not allowed to use children’s play areas, outdoor exercise facilities, and local parks. These restrictions further reduced the option of engaging in PA in the case of densely populated urban centres like Athens. The in-person classes at school were replaced by online courses. Due to the conditions, the content of the PE online courses included cognitive information. PE teachers asked students to fill in the online questionnaires.

The second measurement took place in the middle of February 2021. Schools only opened for 10 days but it was decided to close again due to a significant increase in the ratio of COVID-19 cases. For these 10 days, students attended classes in schools and regularly participated in PE lessons. During these 10 days, structured PA continued not to be allowed and a prerequisite for exercising outdoors was the sending of a specific text message. However, the restrictions that were in force in the previous period concerning the use of play areas, outdoor exercise facilities, and parks were also implemented during this period. Participants answered the online questionnaire in online PE lessons within the next few days after the second lockdown. One of the researchers was available online to answer their questions.

The third measurement was conducted in June 2021 when students had already returned to school classes for almost a month. Lessons in schools were carried out regularly and students participated in PE lessons. Limitations for exercise no longer existed and structured PA started again. However, a decreased number of athletes could participate in organised sports. This time, students filled in the online questionnaires at school as they had been back to regular classes. Thus, the PE teacher gave instructions and answers to their questions.

### 2.4. Data Analysis

Data normal distribution was assessed using the absolute values of skewness and kurtosis due to the large sample size [81]. All data were normally distributed. To explore the construct validity of the subjective vitality scale, a CFA was conducted using the data from the initial measure. These results are presented above in the instruments’ sub-section. Then, descriptive statistics (mean, standard deviation) and Cronbach’s *α* reliability index [82] were calculated. Additionally, to explore the concurrent validity of the examined variables (frequency and amount of MVPA, life satisfaction, and subjective vitality) and the possible relationships among them, correlation analyses were also implemented in all measures. Finally, separate three-way repeated measures analyses of variance (ANOVAs) were conducted to capture possible differences in students’ MVPA, life satisfaction, and subjective vitality among the three measurements due to gender, age, and interaction between gender and age. Post-hoc Sidak test was also used to check for possible differences among the three measurements (time) and groups (independent variables). The level of significance was set at *p* ≤ 0.05.

## 3. Results

### 3.1. Descriptive Statistics, Reliability and Correlation Analyses

Descriptive statistics (means, standard deviations), Cronbach’s *α* reliability index, and absolute values of skewness and kurtosis for all measures are presented below in Table 1. In all measurements, adolescents’ PA levels did not meet the WHO’s recommendations of spending at least 60 min per day in MVPA [6].

Correlation analysis showed significantly positive relations between MVPA variables (days/week, hours/week) in all measurements. Similarly, life satisfaction was positively related to subjective vitality. Positive relations have also emerged between MVPA and well-being variables in all measurements (Table 2).

### 3.2. Differences in Students’ PA and Well-Being Variables among the Three Measurements

Separate three-way repeated measures ANOVAs revealed significant differences in student MVPA frequency (days/week; *F*_2,802_ = 12.503, *p* < 0.001, *η_p_^2^* = 0.03), amount of MVPA (hours/week; *F*_2,808_ = 7.279, *p* = 0.001, *η_p_^2^* = 0.02), life satisfaction (*F*_2,860_ = 72.553, *p* < 0.001, *η_p_^2^* = 0.14) and subjective vitality (*F*_2,860_ = 76.549, *p* < 0.001, *η_p_^2^* = 0.15) among the three measurements (Figure 1; Table 3). More specifically, students reported higher scores at the third measurement (June 2021) in MVPA, life satisfaction and subjective vitality compared to the first (December 2020) and second ones (February 2021).

Regarding the effects of gender and age differences on the frequency of MVPA (days/week), the results revealed no significant interaction between time, gender, and age (*F*_6,802_ = 1.124, *p* = 0.347, *η_p_^2^* = 0.01), no significant interaction between time and gender (*F*_2,802_ = 1.128, *p* = 0.324, *η_p_^2^* = 0.00), and no significant interaction between time and age (*F*_6,802_ = 0.311, *p* = 0.932, *η_p_^2^* = 0.00).

Similarly, regarding the effects of gender and age differences on the amount of MVPA (hours/week), there was no significant interaction between time, gender, and age (*F*_6,808_ = 0.642, *p* = 0.697, *η_p_^2^* = 0.01), no significant interaction between time and gender (*F*_2,808_ = 0.119, *p* = 0.888, *η_p_^2^* = 0.00), and no significant interaction between time and age (*F*_6,808_ = 1.124, *p* = 0.347, *η_p_^2^* = 0.01) (Table 3).

Regarding gender and age differences on life satisfaction, there was no significant interaction between time, gender, and age (*F*_6,860_ = 0.573, *p* = 0.752, *η_p_^2^* = 0.00) and no significant interaction between time and age (*F*_6,860_ = 1.598, *p* = 0.145, *η_p_^2^* = 0.01), but there was significant a significant interaction between time and gender (*F*_2,860_ = 3.536, *p* < 0.05, *η_p_^2^* = 0.01). In analysing this interaction through the Sidak post hoc test, the results showed that boys had higher scores in life satisfaction compared to girls at the first measurement (*F*_1,430_ = 4.646, *p* < 0.05, *η_p_^2^* = 0.01; see Table 3).

Similarly, there was no significant interaction between time, gender, and age (*F*_6,860_ = 1.022, *p* = 0.409, *η_p_^2^* = 0.01), no significant interaction between time and age (*F*_6,860_ = 0.239, *p* = 0.964, *η_p_^2^* = 0.00), but there was a significant interaction between time and gender in subjective vitality (*F*_2,860_ = 5.365, *p* < 0.01, *η_p_^2^* = 0.01). In analysing this interaction through the Sidak post hoc test, the results showed that girls had higher scores in subjective vitality compared to boys at the third measurement (*F*_1,430_ = 5.528, *p* < 0.05, *η_p_^2^* = 0.01; see Table 3).

## 4. Discussion

The main purpose of the present study was to examine differences in adolescents’ MVPA and well-being for six months in which the Greek government had established different protocols to prevent the spread of COVID-19 (closed schools and no structured outdoor activities, opened schools but closed structured outdoor activities, opened schools and structured outdoor activities). The findings indicated that adolescents reported higher levels of MVPA during the third period of the present study in which they could participate in PE lessons and structured or unstructured PA even though there were some limitations (i.e., a small number of participants in structured PA). Additionally, the lowest levels of adolescents’ participation in MVPA were reported during the first phase of the present study in which more severe restrictive measures were implemented. Another finding of this study was that adolescents reported lower MVPA (days/week, hours/week) in all measurements compared to the WHO’s recommendations that children and adolescents should engage in MVPA for at least 60 min daily [6]. These findings are in line with previous studies that found a significant decrease in adolescents’ PA levels during the COVID-19 pandemic [13,59,83]. Thus, it can be argued that the more severe the restrictive measures were, the more decreased students’ MVPA was. The findings of previous studies [12,59,66,67,68,69] also suggested that the social distancing measures (e.g., placing strong restrictions on citizen mobility to necessary and participation in PA) negatively influence adolescents’ participation in PA. Furthermore, students’ higher levels of MVPA during the third phase of the study confirmed the finding of a previous study [84] suggesting that students’ participation in a structured daily program (i.e., a day that includes commuting to school, PE lesson, and outdoor activities or out of school PA) is positively related to PA. An additional factor that may influence adolescents’ participation in PA could be the decrease in adolescents’ social interactions with peers and friends due to the restrictive measures. Peers can significantly influence adolescents’ participation in PA [85].

Additionally, adolescents reported higher levels of well-being (i.e., life satisfaction and subjective vitality) during the third phase of the imposed measures against COVID-19 and lower levels during the first phase. The findings of previous studies suggested that adolescents’ well-being decreased during the COVID-19 restrictive measures [86]. These findings are also in line with previous studies in Greece which found lower levels of children and adolescents’ well-being during the pandemic [13,83,87]. A rational explanation for this could be the severe restrictive measures imposed during the first and second phases of the present study. More specifically, home-schooling during the first phase, the outdoor and indoor sports ban, and the restrictions to outdoor PA or walking during the first and second phases may have led adolescents to report a lower level of well-being in these two phases compared to the third phase of the present study. This assumption stems from the finding of previous studies suggesting that PA affected psychological well-being during the pandemic [13,25,87]. If this is the case, then arguably adolescents during the third phase of the present study reported a higher level of well-being since they could participate in PE lessons and structured and unstructured out of school PA.

An additional factor that may influence adolescent well-being could be that social distancing restrictive measures were established during the first and second phases of the present study. More specifically, during the first phase, adolescents could have in-person contact with their friends and peers for a limited time of the day and on specific occasions. During the second phase, they could also have in-person contact with them during school breaks and activities. Of course, adolescents could be in contact with their peers through social networks during both these phases. In contrast, during the third phase, they could in-person meet their peers and/or friends on any occasion since lifting the restrictive measures had already been relaxed since May. The findings of a previous study supported that peers could be a source of social and emotional support [88] and influence adolescent well-being [89]. Additionally, the findings of a systematic review revealed that in-person contact with peers can affect their well-being. In contrast, the majority of studies revealed that social networks negatively affected the relationship between adolescent’s well-being and their contact with their peers [90].

A second purpose of the present study was to examine possible differences in children’s and adolescents’ PA levels and well-being due to gender and age effects. The findings showed significant differences in children’s and adolescents’ well-being, but not in their PA levels, due to gender. More specifically, girls reported lower scores on life satisfaction compared to boys in the first measurement (closed schools and no structured outdoor activities), while boys reported higher scores on subjective vitality compared to girls in the third measurement (opened schools and structured outdoor activities). No significant differences emerged in PA levels and well-being due to age. These ambiguous findings on adolescents’ well-being during the pandemic are in line with previous research [55,56]. Perhaps, these equivocal findings on well-being or the non-significant differences in PA levels were affected by the restrictive measures, as these were applied to all individuals in the same way. Another reason might be the different methodological tools used to capture adolescents’ PA levels or the different methodological designs selected by the researchers (cross-sectional or longitudinal study). For example, [13,91] used the International Physical Activity Questionnaire (IPAQ; [92] to capture adolescents’ PA levels, while here we used the two items of the WHO’s HBSC study [71]. Furthermore, the present study followed a longitudinal design, while previous studies had a cross-sectional design [13,83,91].

Regarding the weaknesses of the present study, the self-reported measures of PA according to Hallal and his colleagues [8] could be an important limitation because participants had to retrieve from their memory the time they had spent on PA. It is therefore likely that some individuals underestimated or exaggerated their PA behaviour. Additionally, compared to other measurement tools such as the IPAQ [92], which has been widely used in the existing literature to capture adolescents’ PA levels [90], the WHO’s HBSC survey [71], which uses a Likert scale to assess PA behaviour, might not be as accurate (hours instead of 10 min used as unity). In addition, unlike other cross-sectional research [13,93], light PA, such as walking, was not considered in this study. Future studies are encouraged to use motion sensors, such as accelerometers and pedometers, to more accurately capture adolescents’ PA behaviour [94]. Another limitation of the present study is that participants were recruited from a specific high school. However, students from different areas of Athens were enrolled in this specific school. Thus, a representative number of students participated in the study. Furthermore, the present study did not examine the built environment in which students live and exercise that may facilitate or deteriorate adolescents’ participation in PA. The lack of a method to evaluate the participants’ social desirability (e.g., the Marlowe-Crowne Social Desirability Scale) [95] might also be seen as a weakness of this study, as restrictive measures during the pandemic may have worsened their social isolation and loneliness. Future studies are thus advised to employ a social desirability scale to examine any potential gender or age variations that may emerge during a time of more or less rigorous restraints. Finally, another aspect that was not considered in the current study was the participants’ nostalgia for sports activities, despite prior research by Cho and colleagues [96] showing a favourable relationship between this desire to engage in sports and their subjective well-being during the pandemic. Consequently, to investigate potential associations between PA levels and well-being, future research is urged to take into consideration a leisure nostalgia scale [97]. On the contrary, a strength of the present study was that adolescents reported their PA well-being levels for six months, during which different restrictive measures had been applied.

## 5. Conclusions

The COVID-19 restrictions at a national level appear to have influenced adolescents’ PA and well-being. It is well established that both these factors could lead adolescents to positive outcomes. Additionally, PA may act as a catalyst for promoting adolescent well-being. Thus, PA can help adolescents cope with difficulties that the COVID-19 restrictive measures may cause. Hence, if a similar situation arises in the future, policymakers should adopt policies that will provide opportunities to adolescents for participating in out-of-school PA (e.g., open sports fields, more green parks in the cities, and free community-based sports programs for vulnerable adolescents). Policymakers should also avoid the application of severe restrictive measures in young people as this might negatively affect their psychosocial health.

## Figures and Tables

**Figure 1 jfmk-08-00055-f001:**
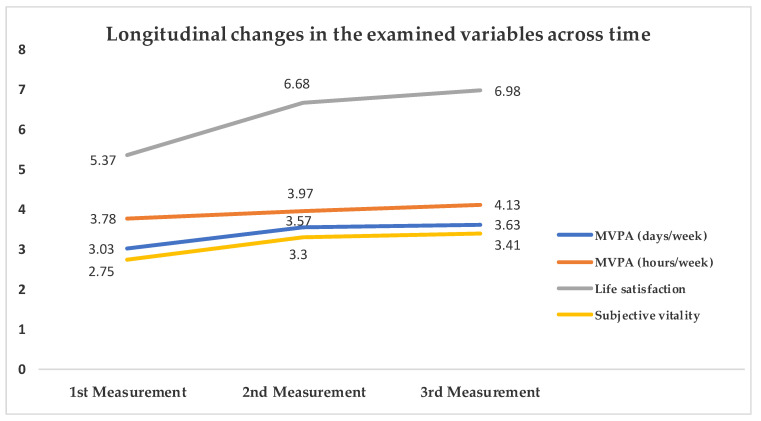
Longitudinal changes in the examined variables across time. MVPA = Moderate to Vigorous Physical Activity. Significant differences emerged across all measurements (*p* ≤ 0.001).

**Table 1 jfmk-08-00055-t001:** Descriptives (means, standard deviations), Cronbach’s *α* reliability index, and absolute values of skewness and kurtosis in all the examined variables.

	1st Measure (December 2020)				2nd Measure (February 2021)				3rd Measure (June 2021)			
Variables	M ± SD	*α*	S	K	M ± SD	*α*	S	K	M ± SD	*α*	S	K
1. MVPA (days/week)	3.03 ± 1.94	-	0.18	−0.71	3.57 ± 1.84	-	0.08	−0.66	3.63 ± 1.83	-	0.10	−0.67
2. MVPA (hours/week)	3.78 ± 1.38	-	−0.45	−0.55	3.97 ± 1.27	-	−0.58	0.12	4.13 ± 1.21	-	−0.70	0.29
3. Life satisfaction	5.37 ± 2.20	-	−0.23	−0.26	6.68 ± 1.95	-	−0.74	0.60	6.98 ± 1.78	-	−0.73	0.87
4. Subjective vitality	2.75 ± 0.74	0.77	0.15	0.36	3.30 ± 0.89	0.87	−0.25	−0.10	3.41 ± 0.88	0.83	−0.40	−0.03

MVPA = Moderate to vigorous intensity physical activity; M = Mean; SD = Standard Deviation; *α* = Reliability index; S = Absolute values of skewness; K = Absolute values of kurtosis.

**Table 2 jfmk-08-00055-t002:** Correlation analysis between the examined variables in all measurements.

	1st Measure (December 2020)				2nd Measure (February 2021)				3rd Measure (June 2021)			
Variables	1	2	3	4	1	2	3	4	1	2	3	4
1. MVPA (days/week)	-				-				-			
2. MVPA (hours/week)	0.63 **	-			0.61 **	-			0.61 **	-		
3. Life satisfaction	0.07	0.04	-		0.15 **	0.15 **	-		0.19 **	0.16 **	-	
4. Subjective vitality	0.24 **	0.19 **	0.51 **	-	0.20 **	0.17 **	0.63 **	-	0.20 **	0.15 **	0.51 **	-

MVPA = Moderate to vigorous intensity physical activity; ** *p* < 0.01.

**Table 3 jfmk-08-00055-t003:** Descriptive statistics and significant differences on the examined variables among the three measurements.

Measurements		1st Measure (December 2020)			2nd Measure (February 2021)			3rd Measure (June 2021)		
	Gender	Boys	Girls	Total	Boys	Girls	Total	Boys	Girls	Total
Variables	Age	M ± SD	M ± SD	M ± SD	M ± SD	M ± SD	M ± SD	M ± SD	M ± SD	M ± SD
MVPA (days/week)	12	2.95 ± 1.92	2.97 ± 1.92	2.96 ± 1.91	3.61 ± 1.95	3.27 ± 1.78	3.42 ± 1.86	3.69 ± 1.85	3.36 ± 1.77	3.51 ± 1.81
	13	3.54 ± 1.97	2.71 ± 1.61	3.08 ± 1.81	3.62 ± 1.78	3.47 ± 1.76	3.53 ± 1.76	3.65 ± 1.73	3.55 ± 1.73	3.59 ± 1.72
	14	3.35 ± 2.15	2.70 ± 1.99	3.03 ± 2.09	4.06 ± 2.03	3.44 ± 1.64	3.76 ± 1.87	4.12 ± 2.09	3.46 ± 1.63	3.80 ± 1.90
	15	3.65 ± 2.06	2.82 ± 1.54	3.32 ± 1.89	3.71 ± 1.90	3.73 ± 1.95	3.71 ± 1.88	3.71 ± 1.90	4.00 ± 2.28	3.82 ± 2.02
	Total	3.30 ± 2.02	2.80 ± 1.82	3.04 ± 1.93 ^a^	3.77 ± 1.93	3.41 ± 1.73	3.58 ± 1.84 ^a^	3.83 ± 1.90	3.48 ± 1.74	3.65 ± 1.83 ^a^
MVPA (hours/week)	12	4.09 ± 1.43	3.93 ± 1.17	4.00 ± 1.29	3.81 ± 1.43	3.96 ± 1.10	3.89 ± 1.26	4.00 ± 1.31	4.09 ± 1.15	4.05 ± 1.22
	13	3.95 ± 1.46	3.47 ± 1.30	3.69 ± 1.39	4.29 ± 1.17	3.88 ± 1.13	4.07 ± 1.16	4.39 ± 1.04	4.02 ± 1.16	4.19 ± 1.12
	14	3.85 ± 1.51	3.44 ± 1.34	3.65 ± 1.44	4.30 ± 1.39	3.53 ± 1.32	3.92 ± 1.41	4.44 ± 1.30	3.77 ± 1.29	4.11 ± 1.34
	15	4.12 ± 1.22	3.45 ± 1.37	3.86 ± 1.30	4.12 ± 1.11	3.73 ± 1.19	3.96 ± 1.14	4.41 ± 0.87	4.00 ± 1.34	4.25 ± 1.08
	Total	3.97 ± 1.44	3.62 ± 1.29	3.79 ± 1.37 ^b^	4.14 ± 1.33	3.80 ± 1.19	3.96 ± 1.27 ^b^	4.29 ± 1.21	3.97 ± 1.21	4.12 ± 1.22 ^b^
Life satisfaction	12	5.79 ± 2.18	4.95 ± 2.33	5.32 ± 2.29	6.63 ± 2.01	7.00 ± 2.11	6.84 ± 2.07	7.40 ± 1.38	7.16 ± 1.98	7.27 ± 1.74
	13	5.34 ± 2.13	5.20 ± 2.30	5.26 ± 2.22	6.73 ± 1.72	6.90 ± 1.90	6.82 ± 1.81	6.97 ± 1.44	7.03 ± 1.84	7.00 ± 1.66
	14	5.79 ± 2.12	5.38 ± 2.11	5.58 ± 2.11	6.41 ± 1.63	6.56 ± 2.03	6.49 ± 1.83	6.77 ± 1.77	6.74 ± 1.84	6.75 ± 1.80
	15	5.47 ± 2.29	4.62 ± 1.81	5.10 ± 2.11	6.29 ± 2.57	6.15 ± 2.19	6.23 ± 2.37	6.47 ± 2.43	6.69 ± 1.80	6.57 ± 2.14
	Total	5.63 ± 2.15 ^c^	5.13 ± 2.23 ^c^	5.37 ± 2.20 ^d^	6.56 ± 1.86	6.79 ± 2.03	6.68 ± 1.95 ^d^	6.99 ± 1.65	6.97 ± 1.89	6.98 ± 1.78 ^d^
Subjective vitality	12	2.88 ± 0.81	2.76 ± 0.74	2.81 ± 0.77	3.39 ± 0.90	3.41 ± 0.88	3.40 ± 0.89	3.44 ± 0.91	3.51 ± 0.87	3.48 ± 0.89
	13	2.74 ± 0.70	2.59 ± 0.70	2.66 ± 0.70	3.10 ± 0.88	3.29 ± 0.91	3.20 ± 0.90	3.19 ± 0.86	3.38 ± 0.92	3.29 ± 0.90
	14	2.95 ± 0.69	2.62 ± 0.75	2.78 ± 0.74	3.24 ± 0.75	3.37 ± 0.94	3.31 ± 0.85	3.30 ± 0.73	3.59 ± 0.92	3.44 ± 0.84
	15	2.67 ± 0.91	2.77 ± 0.46	2.71 ± 0.74	3.20 ± 0.99	3.12 ± 1.03	3.17 ± 0.99	3.22 ± 0.99	3.64 ± 0.84	3.41 ± 0.94
	Total	2.84 ± 0.75	2.67 ± 0.72	2.75 ± 0.74 ^e^	3.24 ± 0.86	3.35 ± 0.91	3.30 ± 0.89 ^e^	3.30 ± 0.85 ^f^	3.50 ± 0.90 ^f^	3.41 ± 0.88 ^e^

M = Mean; SD = Standard Deviation; MVPA = Moderate to vigorous intensity physical activity; ^a, b, d, e^ Significant differences in the examined variables among the three measurements at *p* ≤ 0.001; ^c, f^ Significant differences in the examined variables between boys and girls at *p* < 0.05 in the first and the third measurement, respectively.

## Data Availability

Data is unavailable due to ethical restrictions.

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
