# Peer review of "Adolescents’ Out-of-School Physical Activity Levels and Well-Being during the COVID-19 Restrictions in Greece: A Longitudinal Study"

_jfmk, 2023, doi:10.3390/jfmk8020055_

Round 1
Reviewer 1 Report
Thank you for the opportunity to review this paper. In general, this study the methodologically sound. I appreciated the authors' effort to collect the longitudinal data during the COVID-19 restriction period in Greece.
However, the authors should focus on how the findings of this study can contribute to existing knowledge. To be honest, the answers to two research questions in the introduction (Was adolescent PA reduced? Was adolescent well-being affected?) are very obvious.
Moreover, what the differences between the three measurements can tell us? More discussion should be addressed based on the findings of this study and comparing the findings of this study with other studies.
In addition, as the COVID-19 pandemic is coming to the end, the authors should address how the findings can contribute to practical implications and future research direction.
Good luck.
Author Response
Dear Reviewer,
Thank you very much for your comments. Below, you can find our answers to your valuable comments.
REVIEWER 1
Comments and Suggestions for Authors
Thank you for the opportunity to review this paper. In general, this study the methodologically sound. I appreciated the authors' effort to collect the longitudinal data during the COVID-19 restriction period in Greece.
However, the authors should focus on how the findings of this study can contribute to existing knowledge. To be honest, the answers to two research questions in the introduction (Was adolescent PA reduced? Was adolescent well-being affected?) are very obvious.
Answer: Thank you for your comment. We tried to answer to this comment at the Introduction section. More specifically, we modified the introduction by entering new literature concerning PA and mental health in children and adolescents during the pandemic and by changing the order of some paragraphs. Moreover, we modified the paragraph regarding the purpose and the research hypotheses of the present study.
Moreover, what the differences between the three measurements can tell us?
Answer: Thank you for your comment. We tried to answer to this comment at the Introduction section. More specifically, we modified the introduction by entering new literature concerning PA and mental health in children and adolescents during the pandemic and by changing the order of some paragraphs. Moreover, we changed a lot the paragraph regarding the purpose and the research hypotheses of the present study.
More discussion should be addressed based on the findings of this study and comparing the findings of this study with other studies.
Answer: Thank you for your comment. We revised the Discussion section and compared our findings to previous studies, according to the reviewer's suggestion.
In addition, as the COVID-19 pandemic is coming to the end, the authors should address how the findings can contribute to practical implications and future research direction.
Answer: Thank you for your comment. We revised the Discussion section by entering future study directions and a phrase for the practical implications that policy makers can apply.

Reviewer 2 Report
Thank you for giving me the opportunity to review the manuscript entitled ”Well-being During the COVID-19 Restrictions in Greece: A Longitudinal Study ”
This manuscript aimed to identify the influence of social distancing measures on teenagers’ physical activity.
In general terms, I consider that the article deals with an extremely interesting topic: physical activity, well being and life satisfaction, and it could have an important contribution to the field.
The manuscript is well written and easy to understand, but more transition between parts might make easier to read.
The evaluation of physical activity with only a Lickert scale can be considered as a limitation of this study and is not very precise (hours instead of 10 minutes used as unity) unlike other measurement tools like the IPAQ (International Physical Activity Questionnaire; Craig et al., 2003) that has been extensively used in the literature to assess physical activity behavior (e.g., Đorđić et al., 2022). Indeed, light activity (walking, running) is not considered in the study comparatively to previous studies (e.g., Guo et al., 2021)
Craig CL, Marshall AL, Sjöström M, Bauman AE, Booth ML, Ainsworth BE, et al. International physical activity questionnaire: 12-country reliability and validity. Med Sci Sports Exerc. 2003;35(8):1381–95. 10.1249/01.MSS.0000078924.61453.FB
Guo YF, Liao MQ, Cai WL, Yu XX, Li SN, Ke XY, Tan SX, Luo ZY, Cui YF, Wang Q, Gao XP, Liu J, Liu YH, Zhu S, Zeng FF. Physical activity, screen exposure and sleep among students during the pandemic of COVID-19. Sci Rep. 2021 Apr 20;11(1):8529. doi: 10.1038/s41598-021-88071-4. PMID: 33879822; PMCID: PMC8058040.
Đorđić V, Cvetković M, Popović B, Radanović D, Lazić M, Cvetković B, Andrašić S, Buišić S, Marković M. Physical Activity, Eating Habits and Mental Health during COVID-19 Lockdown Period in Serbian Adolescents. Healthcare (Basel). 2022 Apr 30;10(5):834. doi: 10.3390/healthcare10050834. PMID: 35627970; PMCID: PMC9140993.
In addition, the self-reporting of PA might be another limitation, although its use has been reported in many studies, it is possible that a few participants under- or overestimated their hours of PA. The use of heart rate monitoring and pedometers (e.g., fitness tracker watch), in future studies, should be considered (eventually in the limitation part of the manuscript).
The gender, used as independent variable, is not considered in the introduction nor in the discussion sections of the manuscript.
Suggested references:
Okuyama J, Seto S, Fukuda Y, Funakoshi S, Amae S, Onobe J, Izumi S, Ito K, Imamura F. Mental Health and Physical Activity among Children and Adolescents during the COVID-19 Pandemic. Tohoku J Exp Med. 2021 Mar;253(3):203-215. doi: 10.1620/tjem.253.203. PMID: 33775993.
Solmi M, et al. Physical and mental health impact of COVID-19 on children, adolescents, and their families: The Collaborative Outcomes study on Health and Functioning during Infection Times - Children and Adolescents (COH-FIT-C&A). J Affect Disord. 2022 Feb 15;299:367-376. doi: 10.1016/j.jad.2021.09.090. Epub 2021 Oct 2. PMID: 34606810; PMCID: PMC8486586.
Meade J. Mental Health Effects of the COVID-19 Pandemic on Children and Adolescents: A Review of the Current Research. Pediatr Clin North Am. 2021 Oct;68(5):945-959. doi: 10.1016/j.pcl.2021.05.003. Epub 2021 May 19. PMID: 34538305; PMCID: PMC8445752.
Đorđić V, Cvetković M, Popović B, Radanović D, Lazić M, Cvetković B, Andrašić S, Buišić S, Marković M. Physical Activity, Eating Habits and Mental Health during COVID-19 Lockdown Period in Serbian Adolescents. Healthcare (Basel). 2022 Apr 30;10(5):834. doi: 10.3390/healthcare10050834. PMID: 35627970; PMCID: PMC9140993.
Please find below some suggestions that should be considered in order to improve the quality of the manuscript.
Concerns and suggestions:
Introduction
A definition of PA could be added at the beginning of the introduction.
Pages 2-3, lines 43-44: the sentence should be rephrased and more explicit.
Lines 56-57, I suggest to adding “can also” between “that” and “influence”.
Lines 61-62: a transition is needed between the paragraphs, for better reading.
Hypotheses could be stated at the end of the introduction, in particular concerning gender and age since these factors are used as independent variables (i.e., specified in the data analysis part of the manuscript).
Material Methods
Participants
I suggest rephrasing the first sentence page 3 (line 103) to avoid repetition “Participants”
If available, please give the ethic approval number.
Procedure
Line 151, please remove the full stop after “active-ties.”
Data analysis
Please add in the paragraph that “repeated measures” ANOVAs were conducted for better clarity.
Results
In all this part, please clearly dissociate the fact that the ANOVAs reveal the main effects of each factor as well as the interactions between the factors and that it is the post-hoc tests that reveal where the significant differences lie.
Please start a new paragraph line 223 regarding subjective vitality for better presentation of the results.
Discussion
The results concerning the level and amount of PA should also be discussed with regard to the WHO recommendations.
Limitation
In addition to the limitation mentioned above, as bias can occur with self-report questionnaires the social desirability could have been considered (e.g., Marlowe-Crowne Social Desirability Scale‒ short form C ; Blais, Lachance, & Riddle, 1991; Robin et al., 2018; Verardi et al., 2010)
Verardi, D., Dahourou, J., Ah-Kion, U., Bhowon, C. N., Tseung, D., Amoussou-Yeye, M., ... Rossier, J. (2010). Psychometric properties of the Marlowe-Crowne social desirability scale in eight African countries and Switzerland. Journal of Cross-Cultural Psychology, 41(1), 19–34.
Author Response
Dear Reviewer,
Thank you very much for your comments. Below, you can find our answers to your valuable comments.
REVIEWER 2
Thank you for giving me the opportunity to review the manuscript entitled” Well-being During the COVID-19 Restrictions in Greece: A Longitudinal Study”
This manuscript aimed to identify the influence of social distancing measures on teenagers’ physical activity.
In general terms, I consider that the article deals with an extremely interesting topic: physical activity, well-being and life satisfaction, and it could have an important contribution to the field.
The manuscript is well written and easy to understand, but more transition between parts might make easier to read.
ANSWER: Thank you for your comment. We tried to answer to this comment at the Introduction section. More specifically, we modified the introduction by entering new literature concerning PA and mental health in children and adolescents during the pandemic and by changing the order of some paragraphs. Moreover, we changed a lot the paragraph regarding the purpose and the research hypotheses of the present study.
The evaluation of physical activity with only a Likert scale can be considered as a limitation of this study and is not very precise (hours instead of 10 minutes used as unity) unlike other measurement tools like the IPAQ (International Physical Activity Questionnaire; Craig et al., 2003) that has been extensively used in the literature to assess physical activity behavior (e.g., Đorđić et al., 2022). Indeed, light activity (walking, running) is not considered in the study comparatively to previous studies (e.g., Guo et al., 2021)
Answer: Thank you for your comment. We added a similar paragraph in our Discussion. We also used the proposed references.
Craig CL, Marshall AL, Sjöström M, Bauman AE, Booth ML, Ainsworth BE, et al. International physical activity questionnaire: 12-country reliability and validity. Med Sci Sports Exerc. 2003;35(8):1381–95. 10.1249/01.MSS.0000078924.61453.FB
Guo YF, Liao MQ, Cai WL, Yu XX, Li SN, Ke XY, Tan SX, Luo ZY, Cui YF, Wang Q, Gao XP, Liu J, Liu YH, Zhu S, Zeng FF. Physical activity, screen exposure and sleep among students during the pandemic of COVID-19. Sci Rep. 2021 Apr 20;11(1):8529. doi: 10.1038/s41598-021-88071-4. PMID: 33879822; PMCID: PMC8058040.
Đorđić V, Cvetković M, Popović B, Radanović D, Lazić M, Cvetković B, Andrašić S, Buišić S, Marković M. Physical Activity, Eating Habits and Mental Health during COVID-19 Lockdown Period in Serbian Adolescents. Healthcare (Basel). 2022 Apr 30;10(5):834. doi: 10.3390/healthcare10050834. PMID: 35627970; PMCID: PMC9140993.
In addition, the self-reporting of PA might be another limitation, although its use has been reported in many studies, it is possible that a few participants under- or overestimated their hours of PA. The use of heart rate monitoring and pedometers (e.g., fitness tracker watch), in future studies, should be considered (eventually in the limitation part of the manuscript).
ANSWER: Thank you for your comment. We added a similar paragraph at the Discussion.
The gender, used as independent variable, is not considered in the introduction nor in the discussion sections of the manuscript.
ANSWER: Thank you for your comment. We added a paragraph at the introduction and at the Discussion regarding gender and age differences.
Suggested references:
Okuyama J, Seto S, Fukuda Y, Funakoshi S, Amae S, Onobe J, Izumi S, Ito K, Imamura F. Mental Health and Physical Activity among Children and Adolescents during the COVID-19 Pandemic. Tohoku J Exp Med. 2021 Mar;253(3):203-215. doi: 10.1620/tjem.253.203. PMID: 33775993.
Solmi M, et al. Physical and mental health impact of COVID-19 on children, adolescents, and their families: The Collaborative Outcomes study on Health and Functioning during Infection Times - Children and Adolescents (COH-FIT-C&A). J Affect Disord. 2022 Feb 15;299:367-376. doi: 10.1016/j.jad.2021.09.090. Epub 2021 Oct 2. PMID: 34606810; PMCID: PMC8486586.
Meade J. Mental Health Effects of the COVID-19 Pandemic on Children and Adolescents: A Review of the Current Research. Pediatr Clin North Am. 2021 Oct;68(5):945-959. doi: 10.1016/j.pcl.2021.05.003. Epub 2021 May 19. PMID: 34538305; PMCID: PMC8445752.
Đorđić V, Cvetković M, Popović B, Radanović D, Lazić M, Cvetković B, Andrašić S, Buišić S, Marković M. Physical Activity, Eating Habits and Mental Health during COVID-19 Lockdown Period in Serbian Adolescents. Healthcare (Basel). 2022 Apr 30;10(5):834. doi: 10.3390/healthcare10050834. PMID: 35627970; PMCID: PMC9140993.
ANSWER: Thank you very much for the suggested references. We added most of them at the revised manuscript.
Please find below some suggestions that should be considered in order to improve the quality of the manuscript.
Concerns and suggestions:
Introduction
A definition of PA could be added at the beginning of the introduction.
ANSWER: Thank you for your comment. DONE
Pages 2-3, lines 43-44: the sentence should be rephrased and more explicit.
ANSWER: Thank you for your comment. DONE
Lines 56-57, I suggest to adding “can also” between “that” and “influence”.
ANSWER: Thank you for your comment. DONE
Lines 61-62: a transition is needed between the paragraphs, for better reading.
ANSWER: Thank you for your comment. DONE (we deleted a phrase and modified another one).
Hypotheses could be stated at the end of the introduction, in particular concerning gender and age since these factors are used as independent variables (i.e., specified in the data analysis part of the manuscript).
ANSWER: Thank you for your comment. DONE
Material Methods
Participants
I suggest rephrasing the first sentence page 3 (line 103) to avoid repetition “Participants”
ANSWER: Thank you for your comment. DONE
If available, please give the ethic approval number.
ANSWER: Thank you for your comment. DONE
Procedure
Line 151, please remove the full stop after “active-ties.”
ANSWER: Thank you for your comment. DONE
Data analysis
Please add in the paragraph that “repeated measures” ANOVAs were conducted for better clarity. ANSWER: Thank you for your comment. DONE
Results
In all this part, please clearly dissociate the fact that the ANOVAs reveal the main effects of each factor as well as the interactions between the factors and that it is the post-hoc tests that reveal where the significant differences lie.
ANSWER: Thank you for your comment. We clarified this in the Data analysis Section. Also, in the Results section, we also report Post hoc Sidak test when significant differences lie.
Please start a new paragraph line 223 regarding subjective vitality for better presentation of the results.
ANSWER: Thank you for your comment. DONE
Discussion
The results concerning the level and amount of PA should also be discussed with regard to the WHO recommendations.
Answer: Thank you for your comment. We tried to answer to the above comment by adding a phrase at the Results section and a paragraph at the Discussion.
Limitation
In addition to the limitation mentioned above, as bias can occur with self-report questionnaires the social desirability could have been considered (e.g., Marlowe-Crowne Social Desirability Scale‒ short form C ; Blais, Lachance, & Riddle, 1991; Robin et al., 2018; Verardi et al., 2010)
ANSWER: Thank you for your comment. DONE

Reviewer 3 Report
Congratulations to the authors for the article. It provides a longitudinal description of the relationship between physical activity and well-being in adolescents in Greece.
Minimal improvements are recommended.
The focus of the article should be more concrete.
It would be interesting to include a line graph to see the longitudinal evolution of the main variables, over time.
Author Response
REVIEWER 3
Congratulations to the authors for the article. It provides a longitudinal description of the relationship between physical activity and well-being in adolescents in Greece.
Minimal improvements are recommended.
The focus of the article should be more concrete.
It would be interesting to include a line graph to see the longitudinal evolution of the main variables, over time.
ANSWER: Thank you for your nice words and your comments. We added Figure 1 in the Results to represent in a more demonstrative way the longitudinal evolution of the main variables, over time.

Round 2
Reviewer 1 Report
I can see that the authors put a lot of efforts into the revision. I applaud the improvement of this revised manuscript. I have a last comment on this manuscript. It seems that some up-to-date relevant references are missing. After a quick search, please consider the following references in your study.
Cho, H., Chen, M. Y. K., Kang, H. K., & Chiu, W. (2023). New Times, New Ways: Exploring the Self-Regulation of Sport during the COVID-19 Pandemic and Its Relationship with Nostalgia and Well-Being. Behavioral Sciences, 13(3), 261.
Morres, I. D., Galanis, E., Hatzigeorgiadis, A., Androutsos, O., & Theodorakis, Y. (2021). Physical activity, sedentariness, eating behaviour and well-being during a COVID-19 lockdown period in Greek adolescents. Nutrients, 13(5), 1449.
Jackson, S. B., Stevenson, K. T., Larson, L. R., Peterson, M. N., & Seekamp, E. (2021). Outdoor activity participation improves adolescents’ mental health and well-being during the COVID-19 pandemic. International Journal of Environmental Research and Public Health, 18(5), 2506.
Bates, L. C., Zieff, G., Stanford, K., Moore, J. B., Kerr, Z. Y., Hanson, E. D., ... & Stoner, L. (2020). COVID-19 impact on behaviors across the 24-hour day in children and adolescents: physical activity, sedentary behavior, and sleep. Children, 7(9), 138.
Author Response
Reviewer 1
I can see that the authors put a lot of efforts into the revision. I applaud the improvement of this revised manuscript. I have a last comment on this manuscript. It seems that some up-to-date relevant references are missing. After a quick search, please consider the following references in your study.
Cho, H., Chen, M. Y. K., Kang, H. K., & Chiu, W. (2023). New Times, New Ways: Exploring the Self-Regulation of Sport during the COVID-19 Pandemic and Its Relationship with Nostalgia and Well-Being. Behavioral Sciences, 13(3), 261.
Morres, I. D., Galanis, E., Hatzigeorgiadis, A., Androutsos, O., & Theodorakis, Y. (2021). Physical activity, sedentariness, eating behaviour and well-being during a COVID-19 lockdown period in Greek adolescents. Nutrients, 13(5), 1449.
Jackson, S. B., Stevenson, K. T., Larson, L. R., Peterson, M. N., & Seekamp, E. (2021). Outdoor activity participation improves adolescents’ mental health and well-being during the COVID-19 pandemic. International Journal of Environmental Research and Public Health, 18(5), 2506.
Bates, L. C., Zieff, G., Stanford, K., Moore, J. B., Kerr, Z. Y., Hanson, E. D., ... & Stoner, L. (2020). COVID-19 impact on behaviors across the 24-hour day in children and adolescents: physical activity, sedentary behavior, and sleep. Children, 7(9), 138.
ANSWER: Thank you very much for your comment. We have added the above references! We really appreciate your help to improve this manuscript. We have also improved certain paragraphs in the Discussion.

Reviewer 2 Report
I would like to congratulate the authors for all the changes made which have clearly improved the quality of the manuscript.Author Response
Reviewer 2
I would like to congratulate the authors for all the changes made which have clearly improved the quality of the manuscript.
ANSWER: Thank you very much for your nice words. We really appreciate your help to improve this manuscript.
